# Differences in Technical and Tactical Learning of Football According to the Teaching Methodology: A Study in an Educational Context

**Juan M. García-Ceberino** [1,2,*], **María G. Gamero** [1,2], **Sebastián Feu** [1,2] and **Sergio J. Ibáñez** [1,3]

1   Optimization of Training and Sports Performance Research Group (GOERD), University of Extremadura, 10003 Cáceres, Spain; mgamerob@alumnos.unex.es (M.G.G.); sfeu@unex.es (S.F.); sibanez@unex.es (S.J.I.)
2   Faculty of Education, University of Extremadura, 06006 Badajoz, Spain
3   Faculty of Sports Science, University of Extremadura, 10003 Cáceres, Spain
*   Correspondence: jgarciaxp@alumnos.unex.es; Tel.: +34-627-218-855

**Abstract:** Football performance requires beginning learners to develop both technical skills and tactical awareness. The aim of this study was to examine and contrast the differences in the learning of football across two different teaching methodologies. A total of 35 students, distributed in two class-groups at the fifth-grade level of primary education participated in the study. Each class group participated in just one of the intervention programs (tactical program, $n = 17$; technical program, $n = 18$). The Instrument for the Measurement of Learning and Performance in Football was used to evaluate each student's actions and in relation to specific performance indicators. For each one of the play actions analyzed, the Performance Index of Decision-Making, the Performance Index of Technical Execution, and the Performance Index of Final Results were calculated and these scores were summed to generate the Total Performance Index. The differences in technical and tactical learning between the class-groups were calculated using the Total Performance Index. For this assessment, various statistical tests were used: the Mann–Whitney's U and the Wilcoxon's T (for the non-parametric variables) and the T-test for Independent Samples, as well as the T-test for Related Samples (for the parametric variables). Likewise, a $2 \times 2$ ANOVA was conducted to determine whether the students' previous experience had an effect on the level of learning. The results indicated improvements with both intervention programs; however, the tactical program provided a higher level of learning than the technical program between the assessment tests. The experience of the students had an effect on the play actions of dribbling the ball and in marking the player without the ball. Physical education teachers are recommended to implement comprehensive methods for technical and tactical football teaching at school.

**Keywords:** learning; technical; tactical; experience; tactical games approach; direct instruction

## 1. Introduction

Sport pedagogy is one of the sport science disciplines with the greatest level of application to the educational context. The teachers, teaching and coaching line of study and application is particularly relevant within this area of study [1]. A specific area of focus involves the sport education models that have the purpose of identifying the similarities and differences in the applications of various types of teaching methodologies to sport-based learning. There are two primary areas of instructional approach, which are the teacher-centered approaches (TCAs), and student-centered approaches (SCAs) that direct greater attention to the tactical aspects of instruction [2].

Within the broader framework of TCAs, the method of direct instruction (DI) is the most common. The DI method is oriented to the instructor's teaching behavior, in which the instructor's behavioral processes are considered to be the central focus of the teaching/learning interaction. The instructor, in this case, directs each relevant learning aspect as the students learn and refine their technical skills. The instructor determines the types of learning challenges that will be presented and provides descriptive and prescriptive forms of feedback that are designed to contribute to a better execution of the new motor tasks by the students. In this method, the focus is upon learning the technical skills required by a given activity before proceeding to the development of tactical awareness [3].

In the SCAs design for learning, the tactical games approach (TGA) is one of the methodologies most commonly used. TGA originated as part of the teaching games for understanding the framework proposed by Bunker et al. [4]. The TGA method approaches the teaching/learning process from the perspective of the learner and is designed such that the learner progresses simultaneously through the tactical and technical phases of an activity to develop necessary skills, as well as a broader understanding of the game. In this method, the instructor guides the learners through tactical challenges, including through the use of interrogative feedback [5]. In physical education classes, the use of SCAs for the learning experience is considered to be more beneficial for learners because it allows for greater involvement in the decision-making process, has the potential to generate a deeper understanding of the game [2,6], and can potentially contribute to the physical conditioning of the learners, if appropriately designed [7,8].

Within the physical education context, various studies have examined how students best learn the technical and tactical components of various sport activities, including in the sports of: volleyball [9]; basketball [6,10,11]; and football [12–14]. These studies have provided support for the belief that the student's learning of the tactical aspects of the game tends to be greater when a SCA is utilized. With regard to the development of the technical components of the teaching/learning process, Méndez [15] argued that there is still no conclusive support favoring either TCAs or SCAs. In this regard, da Costa et al. [16] addressed the necessity of directing the attention toward the learning of various specific aspects of the game of football. Some of these concerns related to: (i) the understanding that the majority of game activity for any individual learner will occur in situations when the individual does not have direct contact with the ball; (ii) learners who are currently technically-limited can still play the game with a reasonable level of tactical understanding; and (iii) the development of tactical understanding may contribute to the execution of technical skills. These considerations underlie the need to place greater importance on the learning of the tactical components of various sports at earlier stages of learning. This learning emphasis entails that learners develop the capacity to adapt their play to the different game situations that they will encounter, including considerations such as: temporal and spatial factors; adapting to the pace of the game; identifying personal strengths and weaknesses; and developing awareness of the strengths and weaknesses of the opponent [17].

Regarding the gender of the students, females participate in a better decision-making process and game understanding in invasion sports (e.g., football) than males because they collectively solve the tactical problems. This is because females value the social relationships more than mastering certain technical skills [6,12].

Evaluating student learning in invasion sports has become one of the more problematic challenges faced by physical education instructors [18]. Instructors typically use tests of technical ability in their evaluation of student learning which tends to lead to diminished emphasis on the tactical awareness or decision-making elements of the learning process. In this sense, it is difficult to obtain good, objective data about student learning in these learning environments if tactical learning is not also assessed [19]. Different validated instruments have been generated to assess the learning that is acquired in invasion sports through the systematic observation approach. The Game Performance Assessment Instrument (GPAI) [20] is one such instrument and from it, specific instruments for different invasion sports have been designed and validated, such as for: handball [21], rugby [22], and basketball [23–25].

Within the scientific knowledge base, there are also football instruments that assess the technical and tactical learning of students in actual game situations. Such instruments include the Game Performance Evaluation Tool [26]; the Observation Instrument for Technical and Tactical Actions of the Offensive Phase in Soccer [27]; and the Tool for the Observation and Evaluation of Tactical Offensive Actions in Football [28]. The aforementioned instruments only address player actions during the attack phases of play and thus lack the capacity to assess defensive actions. A more complete instrument called the Instrument for the Measurement of Learning and Performance in Football (IMLPFoot) [29], assesses both the offensive and defensive actions, as well as the actions with or without the ball. This instrument also provides for a more complete assessment of learning in football since it allows for the evaluation of decision making, technical execution and resultant performance.

In order to play the sport of football skillfully, novice learners need to learn both the essential technical skills and develop an understanding of the sport's tactics, such that they will be able to make, and execute, the appropriate decisions during actual game situations [30]. Within the existing knowledge base, few studies have assessed or compared the effectiveness of varied instructional methods on both technical and tactical learning. The purpose of this study was to examine the technical and tactical learning outcomes for the two groups taught with different instructional methods. The hypotheses that were proposed in this study were that: (a) the TGA method would result in higher levels of technical and tactical learning than the DI method; and (b) previous learning experiences would have an influence in the instruction/learning process in sport.

## 2. Materials and Methods

### 2.1. Design of the Study

A quasi-experimental, longitudinal pre/post type of design was utilized in this investigation [31], in which assessments were taken at distinct phases of the learning process for two groups that received different forms of instruction. The study was conducted within a typical educational setting with instruction provided in the sport of football. The groups underwent different instructional protocols with Group A, following a program based on TGA principles and Group B adhering to a program that utilized the DI method. Previous student experience with the game of football was also examined as a possible influence upon learning.

### 2.2. Participants

Participants in the study were 35 beginning learners, of whom 19 (54.3%) were males and 16 (45.7%) were females at the fifth-grade level of primary education. The average age of the participants was 10.63 years ($SD = 0.49$ years). The participants were students in two different groups: 5th grade course, Group A ($n = 17$) received instruction through the Tactical Games Approach Soccer (TGAS) and 5th grade course, Group B ($n = 18$) received Direct Instruction Soccer (DIS) program. When working with indivisible natural class-groups, the selection of each class-group to participate in one instruction program or another was made through random assignment. In this way, the ecological validity of the study was maintained.

An important criterion for the inclusion of each student's result in the study was that students must have participated in at least 80% of the instructional sessions and in both the pre-test and post-test evaluation assessment for their data to be included in the final statistical analyses. The initial sample was comprised of 41 learners but three participants in each group were not included because they did not participate in each of the pre-test and post-test evaluations.

The research was conducted through the Curricular Project Board of the Educational Center following the receipt of approval by the School Council. The study's protocol was consistent with the guidelines for the Helsinki declaration of 1975 and the Organic Law of Protection of Personal Data 15/1999 of December 13 for the protection of personal research data (LOPD) (BOE, 298, December 14, 1999) that have the purpose of maintaining scientific ethics in conducting research with human subjects.

The participation of each individual was subject to the written informed consent of the parents or legal guardians who had previously been informed about any possible risks of participation. The research was conducted through the approval of the Bioethics Committee of the University [Ref: 09/2018].

### 2.3. Variables

The instructional programs were manipulated and served as the independent variable. The two instructional programs were the: (i) Tactical Games Approach for Soccer (TGAS), which is consistent with the TGA methodology; and the (ii) Direct Instruction for Soccer (DIS) program which adheres to the DI method [32]. The consistency of delivery of the two programs was maintained throughout by the same experienced teacher. Subsequently, the delivery of each program was assessed by a panel of experts in order to determine the fidelity to the instructional method and this approach yielded evidence of content validity ($V \geq 0.69$) and internal consistency ($\alpha = 0.970$). A subject expert panel ($n = 13$) with vast experience of teaching methodologies was sought out for assistance by the main researcher [33]. The variable of previous student experience in the game of football (e.g., the practice of football as an out-of-school activity) was assessed to determine the influence of this variable on student learning over the course of this study.

The dependent variables in this study were the 11 distinct game play actions identified in the IMLPFoot instrument. Seven of these game behaviors were offensive actions: (i) dribbling; (ii) shooting; (iii) passing; (iv) controlling the ball; (v) passing and playing (unmarked); (vi) occupying free spaces; and (vii) the recovery of the ball on the attack. Four of the game play behaviors involved defensive actions: (viii) marking the opposing player when the player possessed the ball; (ix) marking of the opposing player when they did not possess the ball; (x) assisting during the transition from attack to defense; and (xi) recovery of the ball as a defender. Within each type of game action, additional assessments were made, including: the decision-making (DM), the technical execution (TE) and the final result (FR). Additional variables were also obtained through these observations, which included: Index of Performance: Decision-Making (IP-DM); Index of Performance: Technical Execution (IP-TE); Index of Performance: Final Result (IP-FR); and the Index of Performance: Total (IP-Total) [24,29]. In invasion sports, the decision-making characteristics are fundamental to success because they involve the capacity to use the information available in any given game situation, and to possess the underlying fundamental game knowledge needed to select an appropriate response from a series of alternatives. Technical execution (TE) is essential because it refers to the ability to appropriately execute the action that has been decided upon [34]. The DM and the TE are essential because they determine the FR of the game action.

### 2.4. Instrument: IMLPFoot

The IMLPFoot instrument was designed so that instructors, coaches and researchers would have a valid and reliable tool to assess learning in the sport of football during the formative years. The instrument has been validated by a panel of individuals with expertise in football and received high estimates of content validity ($V \geq 0.77$), internal consistency ($\alpha = 0.983$) and inter-rater reliability (*Kfree* = considerable or near-perfect) [29].

*System of data coding and retrieval*. The IMLPFoot allows for the simultaneous observation of six players plus both goalkeepers as they play in reduced spaced, small-sided football games. The eleven playing actions, which served as dependent variables, are each assessed through this instrument. For each action that was performed, the components of DM, TE, and FR were assessed with each given a score in relation to the level of performance, where 1 = inadequate, 2 = neutral, and 3 = adequate. In each game of recorded football play, the external coder observed and coded each of the learners' actions when the ball was in play [35], with the beginning and the completion of each ball possession serving as the unit of analysis. Two criteria were established for an action to be assessed as adequate. Thus, if the observed action fulfils the two criteria, it is considered an adequate action; if it only fulfils

one criterion, it is considered a neutral action; and if it fulfils neither of the criteria, it is considered an inadequate action [29].

After the completion of the assessment, the obtained scores for the DM, TE and FR components were summed for each player. A Game Participation score (GP = total number of playing actions in which the learner has participated) was also calculated for each player. Four performance indices were calculated through these scores: (i) IP-DM; (ii) IP-TE; (iii) IP-FR; and (iv) the IP-Total [24,29].

Table 1 displays the formulas that were used for calculating each performance indicator.

**Table 1.** Calculation of performance indices.

| Performance Index | Formula |
|---|---|
| IP-DM | Pts DM/GP |
| IP-TE | Pts TE/GP |
| IP-FR | Pts FR/GP |
| IP-Total | (IP-DM + IP-TE + IP-FR)/3 |

Note: IP-DM = Index of Performance: Decision Making; IP-TE = Index of Performance: Technical Execution; IP-FR = Index of Performance: Final Result; IP-Total = Index of Performance: Total; Pts = Points; DM = Decision Making; TE = Technical Execution; FR = Final Result; GP = Game Participation.

## 2.5. Procedure

Prior to the initiation of this study, various approvals and authorizations were solicited and obtained from the: (i) University's Bioethics Committee; (ii) administrators and physical education instructors at the site where the research was to be completed; (iii) school council of the educational institution involved; and from (iv) the parents or legal guardians of the students who provided informed written consent.

After obtaining the necessary approvals, an initial evaluation was conducted that served as the pre-test in which the students played five games with three players to a team. In this game, two goals of 200 cm were designed with cones but without a goal keeper [30]. The students completed a warm-up session of five minutes prior to the playing in the 3 vs. 3 games. Each of the teams played each of the other teams in this format and there was a two minute rest period and hydration break between games. The games took place on three mini-fields (20 m × 12 m) that were part of a 40 m × 20 m enclosed area with video cameras situated as portrayed in Figure 1. Prior to the pre-test, the learners were provided a questionnaire that requested information relative to: (i) sociodemographic characteristics; (ii) previous football practice in non-school settings; and (iii) years of previous football practice. The questions were intended to provide the information that would allow for the fair balancing of teams. The teams were mixed by gender and they had players with experience and inexperience.

The TGAS and DIS class-groups were then formed for the nine class sessions devoted to learning football. Each class session included four learning tasks of 10 min duration. The sessions were designed to be increasingly challenging, beginning with simple tasks (1 vs. 0, 2 vs. 0, 1 vs. 1 … ) before advancing to more complex challenges (3 vs. 3, 4 vs. 4, 5 vs. 5 … ). The TGAS program was structured in relation to the tasks that were designed to develop tactical knowledge in which student decision-making processes would be essential for positive outcomes. The DIS program was structured according to the development and refinement of the specific technical skills that would be integrated into regular game play once they were habitual or automatic [32]. Upon the completion of the daily lessons using either the TGAS or DIS instruction, a post-test evaluation was conducted and the participants returned to playing 3 vs. 3 games in which the structure, game length and rest periods were the same as during the pre-test. The composition of the teams was also the same as in the pre-test. All of the game play that comprised the pre-test and post-assessments were video recorded using three video cameras for the analysis. The teams were differentiated by colored jerseys to facilitate the video analysis.

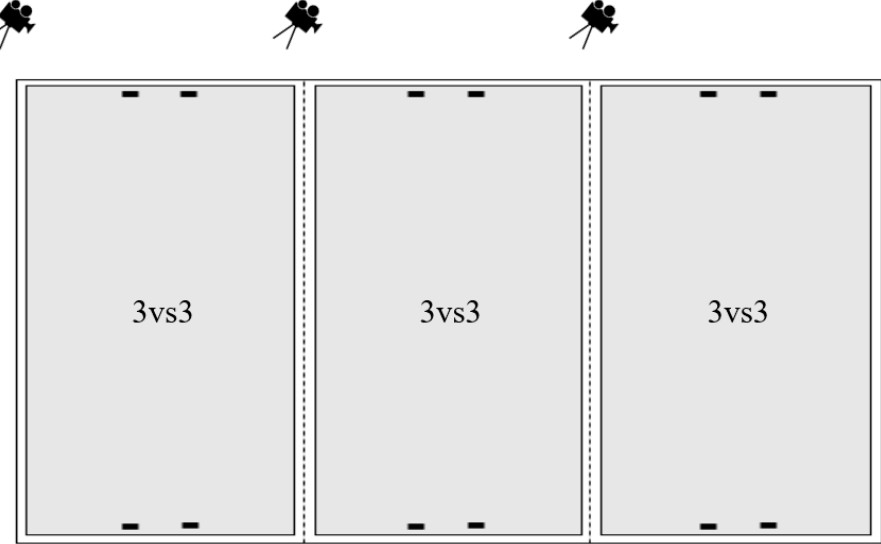

**Figure 1.** Organization of the mini-games.

At the final step, an expert coder of formative football assessed and coded each of the players' actions across the 60 total games (*n* pre = 30; *n* post = 30) through the use of the IMLPFoot instrument [29]. Intra-rater reliability was computed using Cohen's Kappa Index [36] through the evaluation of a randomly selected game that had also been assessed and coded fifteen days previously by the same evaluator. The results obtained from this analysis were examined with Cohen's Kappa Index and these findings revealed acceptable levels of consistency across time [37]. Moderate consistency was found across time for the marking of the player without the ball ($K = 0.60$) and high levels of consistency were obtained for: passing ($K = 0.70$); passing and playing (unmarked) ($K = 0.79$); occupying free spaces without the ball ($K = 0.62$); and marking of the player with the ball ($K = 0.75$). Higher Kappa values were obtained for: dribbling ($K = 1.00$); shooting ($K = 0.86$); ball control ($K = 0.82$); recovery of the ball by the defenders ($K = 1.00$); assisting on the transition from attach to defense ($K = 1.00$); and recovery of the ball when defending ($K = 1.00$).

*2.6. Statistical Analysis*

Statistical assumptions were assessed relative to normality, homogeneity of variance and randomness for the purpose of selecting the appropriate statistical approach to test the hypotheses [38]. These results indicated that parametric tests were appropriate for examining hypotheses that used data for the variables of: passing and playing (unmarked) and for marking the player with the ball. Nonparametric tests were conducted to contrast group differences across the remaining paired variables. Descriptive data was also obtained for each group in order to obtain means and standard deviations for the performance indicators for the pre-test and post-test assessments.

Subsequently, to contrast pre-test differences in initial performance between learners in both groups, the *Mann–Whitney's U test* was employed to test the group differences from the nonparametric data variables, and an *Independent Samples T-test* was used for the data for variables that had met parametric assumptions. The same statistical process was conducted to contrast the differences in final performance between the groups on these variables at the time of the post-test [39]. These statistics are presented in Figure 2.

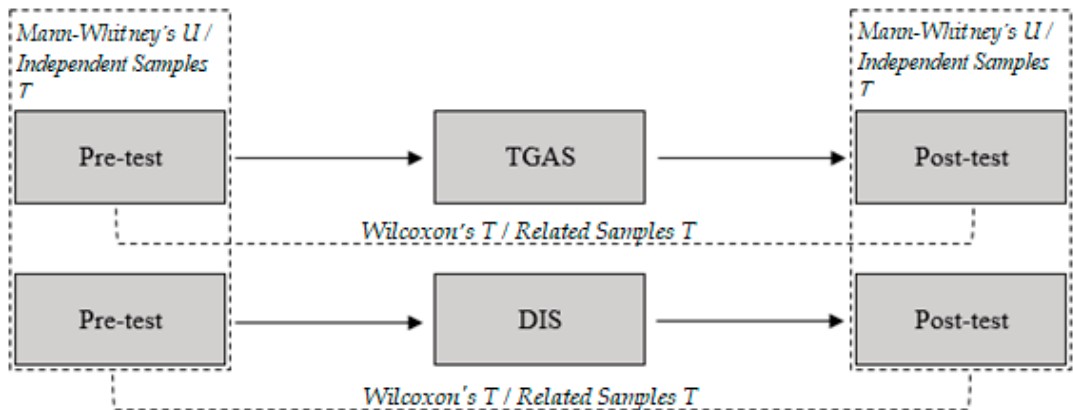

**Figure 2.** Statistical tests of the differences within and between groups.

In order to test for the differences in learning between each class-groups between the pre-test and post-test the *Wilcoxon's T statistic* (for data from nonparametric variables) and the *Related Samples T-test* for parametric data were computed [39]. These results are also provided in Figure 2. To examine the possible effect of the participants' previous experience on the amount of learning that occurred over the course of the study, a $2 \times 2$ RM ANOVA was conducted. To conduct this analysis, it was necessary to first transform the nonparametric variable data to parametric data [40].

All of the inferential analyses were conducted on the outcome variable of IP-Total, which was comprised of the summed scores of the performance values for IP-DM, IP-TE and IP-FR (see Table 1). A significance level of $p < 0.05$ was considered. All statistical analyses were conducted using SPSS 21.0 (IBM Corp. Released 2012. IBM SPSS Statistics for Windows, Version 21, IBM Corp, Armonk, NY, USA).

At the final step, an effect size was calculated using *Cohen's d statistic* and *eta partial squares* values [41]. These findings were interpreted in relation to the ranges that have been established previously [36], where <0.000 (negative effect); 0.00–0.199 (no effect); 0.200–0.499 (small effect); 0.500–0.799 (moderate effect); and 0.800–1.000 (large effect).

## 3. Results

Table 2 provides the descriptive results relative to the performance assessments of each of the game play actions in the pre-test and post-test assessments of the TGAS and DIS intervention groups. In relation to IP-Total, it can be observed that there was an improvement in the levels of learning in each of the learning contexts across the intervention.

The between-group comparisons are provided in Table 3. No significant differences emerged in the game play actions between the two groups except for the IP-Total scores for the actions of recovery of the ball for both attackers and defenders (pre-test). Likewise, there were no significant differences in the player performance in the component skill dimensions between the TGAS and DIS programs at the post-test.

Table 4 provides the results for the differences between each class-group of the total performance scores from the pre-test to post-test. These results show that the TGAS learners had a general tendency toward a better performance than did the DIS learners.

Finally, the effects of the participants' previous experience on the level of learning of the 11 game play actions is presented in Table 5. These findings indicate that the learners' previous experience playing football had effects for the skills of dribbling and marking the player without the ball.

**Table 2.** Descriptive data for the pre-test and post-test assessment.

| Play Actions | | Indices | Pre-Test | | Post-Test | | Pre-Test | | Post-Test | |
|---|---|---|---|---|---|---|---|---|---|---|
| | | | *M* | *SD* | *M* | *SD* | *M* | *SD* | *M* | *SD* |
| Attacking actions | Dribbling | IP-DM | 2.80 | 0.25 | 2.68 | 0.58 | 2.85 | 0.13 | 2.99 | 0.03 |
| | | IP-TE | 1.81 | 0.33 | 1.71 | 0.47 | 1.82 | 0.30 | 2.02 | 0.21 |
| | | IP-FR | 2.31 | 0.64 | 2.11 | 0.79 | 2.44 | 0.37 | 2.25 | 0.60 |
| | | IP-Total | 1.49 | 1.16 | 2.17 | 0.48 | 1.45 | 1.20 | 1.61 | 1.19 |
| | Shooting | IP-DM | 2.32 | 0.30 | 2.43 | 0.38 | 2.22 | 0.19 | 2.23 | 0.45 |
| | | IP-TE | 2.04 | 0.49 | 2.26 | 0.33 | 2.23 | 0.57 | 2.23 | 0.53 |
| | | IP-FR | 1.51 | 0.49 | 1.56 | 0.58 | 1.45 | 0.44 | 1.33 | 0.32 |
| | | IP-Total | 1.61 | 0.83 | 2.08 | 0.34 | 1.64 | 0.80 | 1.82 | 0.55 |
| | Passing | IP-DM | 2.55 | 0.26 | 2.54 | 0.19 | 2.60 | 0.25 | 2.70 | 0.26 |
| | | IP-TE | 1.83 | 0.20 | 1.96 | 0.20 | 1.85 | 0.22 | 1.98 | 0.21 |
| | | IP-FR | 2.46 | 0.27 | 2.49 | 0.28 | 2.50 | 0.38 | 2.55 | 0.35 |
| | | IP-Total | 2.28 | 0.20 | 2.33 | 0.19 | 2.31 | 0.26 | 2.41 | 0.24 |
| | Controlling the ball | IP-DM | 2.32 | 0.47 | 2.46 | 0.45 | 2.59 | 0.31 | 2.57 | 0.42 |
| | | IP-TE | 2.27 | 0.55 | 2.37 | 0.52 | 2.52 | 0.48 | 2.50 | 0.58 |
| | | IP-FR | 2.34 | 0.52 | 2.52 | 0.54 | 2.61 | 0.49 | 2.56 | 0.57 |
| | | IP-Total | 2.31 | 0.50 | 2.45 | 0.50 | 2.29 | 0.92 | 2.54 | 0.50 |
| | Passing and playing (unmarked) | IP-DM | 2.13 | 0.50 | 2.21 | 0.28 | 2.07 | 0.57 | 2.41 | 0.39 |
| | | IP-TE | 2.17 | 0.22 | 2.26 | 0.18 | 2.16 | 0.28 | 2.14 | 0.22 |
| | | IP-FR | 1.93 | 0.39 | 2.01 | 0.30 | 1.82 | 0.50 | 2.04 | 0.29 |
| | | IP-Total | 2.08 | 0.36 | 2.16 | 0.24 | 2.03 | 0.43 | 2.20 | 0.26 |
| | Occupying free spaces | IP-DM | 2.54 | 0.20 | 2.45 | 0.18 | 2.53 | 0.22 | 2.49 | 0.18 |
| | | IP-TE | 2.02 | 0.18 | 2.08 | 0.13 | 2.09 | 0.96 | 2.16 | 0.18 |
| | | IP-FR | 2.64 | 0.23 | 2.64 | 0.20 | 2.65 | 0.20 | 2.65 | 0.16 |
| | | IP-Total | 2.40 | 0.16 | 2.39 | 0.13 | 2.42 | 0.14 | 2.43 | 0.09 |
| | Ball recovery (by the attackers) | IP-DM | 1.51 | 0.64 | 1.85 | 0.57 | 1.61 | 0.54 | 1.67 | 0.64 |
| | | IP-TE | 1.51 | 0.64 | 1.82 | 0.56 | 1.61 | 0.54 | 1.67 | 0.64 |
| | | IP-FR | 1.42 | 0.59 | 1.64 | 0.44 | 1.50 | 0.40 | 1.53 | 0.46 |
| | | IP-Total | 1.05 | 0.87 | 1.77 | 0.51 | 1.57 | 0.49 | 1.62 | 0.57 |
| Defending actions | Marking the player with the ball | IP-DM | 2.04 | 0.28 | 2.17 | 0.31 | 2.08 | 0.23 | 2.22 | 0.30 |
| | | IP-TE | 2.06 | 0.30 | 2.09 | 0.39 | 2.12 | 0.27 | 2.23 | 0.32 |
| | | IP-FR | 1.53 | 0.32 | 1.74 | 0.30 | 1.63 | 0.19 | 1.63 | 0.22 |
| | | IP-Total | 1.88 | 0.27 | 2.00 | 0.32 | 1.94 | 0.21 | 2.03 | 0.27 |
| | Marking the player without the ball | IP-DM | 1.82 | 0.28 | 1.97 | 0.32 | 1.93 | 0.22 | 1.96 | 0.32 |
| | | IP-TE | 1.79 | 0.21 | 1.87 | 0.28 | 1.90 | 0.24 | 1.92 | 0.24 |
| | | IP-FR | 1.24 | 0.20 | 1.31 | 0.20 | 1.30 | 0.19 | 1.36 | 0.25 |
| | | IP-Total | 1.62 | 0.20 | 1.72 | 0.25 | 1.71 | 0.18 | 1.75 | 0.25 |
| | Assisting/defensive change | IP-DM | 2.14 | 0.55 | 2.08 | 0.39 | 2.17 | 0.22 | 2.27 | 0.25 |
| | | IP-TE | 2.29 | 0.72 | 2.54 | 0.54 | 2.86 | 0.28 | 2.80 | 0.23 |
| | | IP-FR | 1.97 | 0.68 | 2.17 | 0.42 | 2.35 | 0.48 | 2.23 | 0.31 |
| | | IP-Total | 2.13 | 0.58 | 2.26 | 0.39 | 2.46 | 0.23 | 2.43 | 0.16 |
| | Ball recovery (by the defenders) | IP-DM | 1.79 | 0.86 | 1.72 | 0.63 | 1.76 | 0.38 | 1.70 | 0.52 |
| | | IP-TE | 1.79 | 0.86 | 1.71 | 0.63 | 1.76 | 0.38 | 1.70 | 0.52 |
| | | IP-FR | 1.52 | 0.59 | 1.57 | 0.58 | 1.60 | 0.35 | 1.61 | 0.44 |
| | | IP-Total | 1.10 | 1.03 | 1.67 | 0.61 | 1.70 | 0.35 | 1.57 | 0.62 |

Note: *M* = Mean; *SD* = Standard Deviation; IP-DM = Index of Performance: Decision Making; IP-TE = Index of Performance: Technical Execution; IP-FR = Index of Performance Final Result; IP-Total = Index of Performance: Total.

**Table 3.** Differences in pre-test and post-test scores relative to the teaching methodology.

| Indexes-Play Actions | | Pre-Test | | | Post-Test | | |
|---|---|---|---|---|---|---|---|
| | | U/t | p | d | U/t | p | d |
| Attack | IP-Total-Dribbling | 152,500 [a] | 0.99 | 0.01 | 104,000 [a] | 0.60 | 0.19 |
| | IP-Total-Shooting | 150,000 [a] | 0.92 | 0.03 | 110,500 [a] | 0.25 | 0.40 |
| | IP-Total-Passing | 128,500 [a] | 0.42 | 0.28 | 114,500 [a] | 0.20 | 0.44 |
| | IP-Total-Controlling the ball | 130,500 [a] | 0.46 | 0.25 | 129,000 [a] | 0.43 | 0.27 |
| | IP-Total-Passing and playing (unmarked) | 0.342 [b] | 0.73 | 0.12 | −0.426 [b] | 0.67 | 0.14 |
| | IP-Total-Occupying free spaces | 144,000 [a] | 0.77 | 0.10 | 123,000 [a] | 0.32 | 0.34 |
| | IP-Total-Ball recovery (by the attackers) | 88,500 [a] | 0.03 * | 0.77 | 129,500 [a] | 0.44 | 0.26 |
| Defense | IP-Total-MPwB | −0.808 [b] | 0.42 | 0.27 | −0.242 [b] | 0.81 | 0.08 |
| | IP-Total-MPwoB | 103,500 [a] | 0.10 | 0.57 | 140,000 [a] | 0.67 | 0.14 |
| | IP-Total-Assisting/defensive change | 94,000 [a] | 0.05 | 0.70 | 99,500 [a] | 0.12 | 0.55 |
| | IP-Total-Ball recovery (by the defenders) | 89,000 [a] | 0.03 * | 0.76 | 147,500 [a] | 0.85 | 0.06 |

Note: IP-Total = Index of Performance: Total; MPwB = Marking the Player with the Ball; MPwoB = Marking the Player without the Ball; [a] Mann–Whitney's U; [b] Independent Samples T. * $p < 0.05$.

**Table 4.** Pre-test and post-test statistical differences by class-group.

| Indexes | | TGAS (Pre-Post) | | | | DIS (Pre-Post) | | | |
|---|---|---|---|---|---|---|---|---|---|
| | | Z/t | R | p | d | Z/t | R | p | d |
| Attack | IP-Total-Dribbling | −2.433 [a] | + | 0.01 * | 1.83 | −1.177 [a] | + | 0.24 | 0.58 |
| | IP-Total-Shooting | −1.448 [a] | + | 0.15 | 0.78 | −0.828 [a] | c | 0.41 | 0.40 |
| | IP-Total-Passing | −0.592 [a] | + | 0.55 | 0.29 | −0.980 [a] | + | 0.33 | 0.47 |
| | IP-Total-Controlling the ball | −1.728 [a] | + | 0.08 | 0.92 | −0.327 [a] | + | 0.74 | 0.15 |
| | IP-Total-Passing and playing | −0.777 [b] | − | 0.45 | 0.27 | −1.570 [b] | c | 0.13 | 0.45 |
| | IP-Total-Occupying free spaces | −0.142 [a] | + | 0.89 | 0.07 | −0.849 [a] | + | 0.40 | 0.41 |
| | IP-Total-Ball recovery (by attackers) | −2.444 [a] | + | 0.01 * | 1.47 | −0.310 [a] | c | 0.76 | 0.15 |
| Defense | IP-Total-MPwB | −2.123 [b] | + | 0.05 | 0.41 | −1.600 [b] | + | 0.13 | 0.33 |
| | IP-Total-MPwoB | −2.012 [a] | + | 0.04 * | 1.12 | −1.067 [a] | c | 0.29 | 0.52 |
| | IP-Total-Assisting/defensive change | −0.104 [a] | − | 0.92 | 0.05 | −0.569 [a] | − | 0.57 | 0.27 |
| | IP-Total-Ball recovery (by defenders) | −1.818 [a] | + | 0.07 | 0.98 | −0.554 [a] | − | 0.55 | 0.26 |

Note: TGAS = Tactical Games Approach Soccer program; DIS = Direct Instruction Soccer program; IP-Total = Index of Performance: Total; MPwB = Marking the Player with the Ball; MPwoB = Marking the Player without the Ball; [a] Wilcoxon's T; [b] related samples T; R = range (+), positive tendency for range; R = range (c), constant for range; R = range (−), negative tendency for intervention program. * $p < 0.05$.

**Table 5.** Effects of previous experience on the level of learning.

| Indexes-Play Actions | | F | p | Eta |
|---|---|---|---|---|
| Attack | IP-Total-Dribbling | 9.398 | 0.00 * | 0.26 |
| | IP-Total-Shooting | 0.427 | 0.52 | 0.01 |
| | IP-Total-Passing | 0.104 | 0.75 | 0.00 |
| | IP-Total-Controlling the ball | 0.187 | 0.67 | 0.01 |
| | IP-Total-Passing and playing (unmarked) | 0.668 | 0.42 | 0.21 |
| | IP-Total-Occupying free spaces | 0.935 | 0.34 | 0.03 |
| | IP-Total-Ball recovery (by the attackers) | 1.926 | 0.17 | 0.06 |
| Defense | IP-Total-MPwB | 1.295 | 0.26 | 0.04 |
| | IP-Total-MPwoB | 6.055 | 0.02 * | 0.16 |
| | IP-Total-Assisting/defensive change | 0.114 | 0.74 | 0.00 |
| | IP-Total-Ball recovery (by the defenders) | 0.000 | 0.98 | 0.00 |

Note: IP-Total = Index of Performance: Total; MPwB = Marking the Player with the Ball; MPwoB = Marking the Player without the Ball. * $p < 0.05$.

### 4. Discussion

Football play requires the learning of technical skills and tactics awareness by the learners. The purpose of this study was to contrast the technical and tactical learning rates of the students participating in two different instructional situations using a pre-test/post-test design. The two groups in this study received either the TGAS or DIS forms of instruction. In addition, the influence of students' previous experience on learning outcomes was assessed. The results revealed that both groups demonstrated improvements in both the technical and tactical learning outcomes. Nonetheless, the students that were instructed using the TGAS program demonstrated greater change from the pre-test to the post-test than did the learners who were in the instructional setting that used the DIS program. In addition, previous experience by learners was found to be influential upon learning.

*4.1. Evaluation of Learning*

The learning obtained by each class-group was assessed through a variety of skill indicators in the pre-test and post-test periods. For the inferential analyses, the variable of IP-Total was generated through the combination of scores on the individual performance components that included the IP-DM, IP-TE and IP-RF that comprise the IMLPFoot [29]. This evaluation instrument allows for the collection of objective data relative to the technical and tactical learning of the learners [19]. An additional instrument that has been designed and validated for the assessment of performance in invasion-type games (basketball) also employed the indicators of the game participation and various performance outcome measures [24]. The assessment of learning over the course of the interventions in school should include the evaluation of decision-making processes and technical skill performance improvements in order to provide a more comprehensive understanding of the amount of learning that occurred.

*4.2. Between-group Differences*

The intervention programs contributed to skill learning in both classes. There were no significant differences between the groups at the initial pre-test, nor at the post-test that was completed at the conclusion of the intervention, except in the ball recovery by the attackers and defenders (pre-test) in favor of the DI method. Previous researchers had found the TGA method to be more effective in contributing to the learning of invasion sports in the school context [6,10,42]. It is possible that the type of sport may have an effect upon the development of knowledge in the learners. It is necessary that instructors also consider the technical and tactical skill demands of the activities which they wish to instruct when making decisions about instructional methods. In this study, there was no definitive evidence that one instructional method was better than the other in contributing to learning outcomes, and further research is needed to better understand the learning process of beginning learners in this sport.

*4.3. Within-Group Differences*

Each of the groups demonstrated growth in learning from pre-test to post-test. The TGAS group tended to achieve higher range values for technical and tactical learning in comparison to the DIS. The students in the TGAS group demonstrated significant improvement in: dribbling, ball recovery by the attackers and the off-ball marking of opponents. On the other hand, the students in the DIS class did not demonstrate significant improvement across any of the 11 play actions. These differences may be attributable to a greater interest of beginners to participate in game play. Previous research has found that SCA-based programs contribute to greater engagement in decision-making and level of understanding of the game [12–14]. The SCAs cause a higher perception of competence [43]. Likewise, this perceived competence favors the students' intrinsic motivation (e.g., satisfaction or fun while playing sports) because they develop the tactical awareness using the game [44]. Greater perceived competence is associated to a higher students' intention to be physically active [45], so that the SCAs

promote a healthy lifestyle [46]. In addition, these teaching approaches can contribute to the greater physical conditioning of students [7,8].

The TGAS program is presumed to be desirable because it can contribute to technical skill execution in actual game situations, thus enabling students to attain beneficial results throughout the learning process. The improvements in technical skill learning are attributable to the efforts to direct attention to tactical awareness that underlie the TGA method [47]. Various investigators who have assessed actual game play have encountered a more favorable learning of technical aspects of the game when teaching methods are linked to game understanding [6,10,11]. Consequently, it is appropriate to enhance instruction in the technical learning components through SCAs that allow students to better understand the purpose of technical skills (what, how and when) in in the larger picture of game play.

### 4.4. Effect of Previous Experience

Previous football experience was related to the amount of acquired learning that occurred through the intervention. In this respect, the results revealed that learners with less previous football experience improved more than the more experienced learners on the skills of dribbling and marking opposing players who were off ball. Furthermore, the learners with less initial football ability improved more in their tactical awareness than did those with greater initial ability [30,48]. It can be presumed that a relatively greater amount of learning is likely to occur in invasion sports for those individuals who learn in environments that foster more active student engagement [49].

### 4.5. Future Research and Limitations

In the context of physical education, it is essential that learners know how to use the information that is available in their environment to make appropriate decisions and also to develop the capacity to transfer this knowledge to other sports and contexts using their internalized knowledge [14,50]. In physical education classes, it is advisable that students learn to play invasion sports using an SCAs and it can be assumed that this learning will transfer to other sports in which they participate within the curriculum [11].

Given the anticipated beneficial effects of an SCA-structured learning environment, relative to a TCA-structured learning environment, it is recommend that instructors utilize the SCA form of instruction in teaching invasion sports such as football in physical education classes [2,9]. Future studies in this area might also wish to examine the role of gender in the teaching/learning process. Various researchers have found that females engage in a better decision-making process and game understanding than males in these types of learning situations because they collectively solve tactical problems [6,12]. Another consideration to keep in mind is that the quality of training and preparation of instructors may be influential in shaping student learning outcomes in these types of learning activities. In this regard, researchers [51] have demonstrated that instructors with better training in teaching tactical games have better learning results in Physical Education classes. One of the limitations of the present study was the sample size and future research can also benefit from larger samples of learners.

## 5. Conclusions

The implementation of the intervention programs that included two different methods to the teaching/learning process in an invasion sport in the educational context resulted in improvements in learning for the participants. Although significant overall differences did not emerge as a result of the contrasting instructional styles that centered on the instructor (DI) or the student (TGA), the students that participated in the more comprehensive instruction improved to a greater extent in their learning of technical components and tactical awareness. The non-direct teaching method resulted in greater levels of learning in this invasion sport for individuals with less personal experience and consequently, the non-direct style would seem to be ideal at the beginning phases of instruction.

In conducting research along this line, instruments that include assessments of decision-making processes, technical execution and overall efficacy are beneficial and needed. Through the use of a

more comprehensive assessment system, a better understanding can be gained about the learning of the component skills of the sport and in relation to overall performance. Given the known beneficial characteristics of the SCAs learning environment, its use is recommended in physical education.

**Author Contributions:** Conceptualization, J.M.G.-C., S.F. and S.J.I. methodology, J.M.G.-C., S.F. and S.J.I. formal analysis, J.M.G.-C. reviewers, M.G.G., S.F. and S.J.I. writing—original draft preparation, J.M.G.-C. writing—review and editing, M.G.G., S.F. and S.J.I. visualization, J.M.G.-C. supervision, M.G.G., S.F. and S.J.I. All authors have read and agreed to the published version of the manuscript.

**Funding:** This study has been partially subsidized by the Aid for Research Groups (GR18170) from the Regional Government of Extremadura (Department of Employment, Companies and Innovation), with a contribution from the European Union from the European Funds for Regional Development.

**Acknowledgments:** The authors thank the Rodeo school, the physical education teachers, and the students for participating in the research.

**Conflicts of Interest:** The authors declare no conflict of interest.

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
