# Peer review of "Differences in Technical and Tactical Learning of Football According to the Teaching Methodology: A Study in an Educational Context"

_sustainability, doi:10.3390/su12166554_

Round 1
Reviewer 1 Report
Title: a suggestion is done à Differences in technical and tactical learning according to the teaching methodology: a study in an educational context in football
Abstract: In the beginning, the authors introduce this sentence: “Football performance requires beginning learners to develop both technical skills and tactical awareness”. I think this idea is not related to the aim of the study (methodology) and also points out a “special” topic related to this question: it is necessary to start playing football as soon as possible in order to achieve greater performances?
According to sample size, a mistake is found: 37 is not 17+18 (group 1 + group 2). The authors write 35 in the method. Please, modify it!
Moreover, from lines 19 to 22 we only read about instruments. Maybe the authors can reduce this text and add at the end of the abstract the contribution (usefulness) of the main findings. I think this is more interesting for readers!
Keywords: I would like to suggest including “learning” (maybe other keywords could be removed)
Introduction
Many information about instruments of evaluation is offered, however, a lack of information is showed about teaching/learning methodologies. I think the introduction should have a great modification in order to offer more insight into learning strategies focused on technical or tactical methods. This change implies a great modification of this section. Moreover, I think the methodological information (pre-post tests) is not necessary here. The reader can find all the details in the next sections.
Method
Please, give more information about the panel of experts (experience, level, number…). I think it is a great contribution to this research study.
Please, explain with more detail what the authors consider as “student experience” (lines 146-147). Experience in deliberate practice? in competition?
Lines 180-187: I think this information could be set close to the definitions of the variables (after line 165). Moreover, I suggest including more details related to the categories: 1 = inadequate, 2 = neutral, and 3 = adequate. I encourage explaining it according to the different components (DM, TE, and FR).
Regarding the information shown in lines 201-204, please, give more details about the information requested to the students and how this information was managed in order to create properly balanced teams
Lines 208. Four learning situations were used, but how many were “simple” vs “complex” tasks? Please, also offer a greater explanation about the key instructions (feed-forward) used by the teacher before each exercise. The information about TGAS and DIS sessions is excessively general (like an introduction). Please, give more details about the task constraints, specific goals for sessions, etc. (no only about the general aims).
In this “procedure” sub-section, please, offer more details about the pre-post test situation. Finally, it was done an inter-reliability process?
Concerning the 2 “statistical analysis” sub-section, please, add the level of significance used in the tests.
Results
Nice! Congrats for this good section.
Discussion
Lines 313-onwards. According to table 3, a few statistical differences were found between groups in the pre-test. This fact should be commented on.
The rest of the discussion is nice and well structured
Congratulations for this nice study!
Author Response
First of all, we would like to express our gratitude to reviewer 1 for the time in reviewing our manuscript and for providing us comments helpful to improve this manuscript quality. We have found suggestions very constructive and have answered their concerns point by point.
--------------------------------
All manuscript
- A native translator performed a grammatical revision of the manuscript.
- All corrections were marked in red.
--------------------------------
Reviewer’ note: It is necessary to start playing football as soon as possible in order to achieve greater performances?
Authors’ response: In Spain, the teaching of invasion sports (e.g. football) at school begins at the fifth level of Elementary Education. Students who participated in the study meet at this level.
--------------------------------
Reviewer’ note: According to sample size, a mistake is found: 37 is not 17+18 (group 1 + group 2). The authors write 35 in the method. Please, modify it!
Authors’ response: The sample was corrected (Line 14).
--------------------------------
Reviewer’ note: Moreover, from lines 19 to 22 we only read about instruments. Maybe the authors can reduce this text and add at the end of the abstract the contribution (usefulness) of the main findings. I think this is more interesting for readers!
Authors’ response: These lines mention the variables studied. A practical contribution was added (Line 30 to 31).
--------------------------------
Reviewer’ note: I would like to suggest including “learning” (maybe other keywords could be removed).
Authors’ response: Added "learning" as a keyword and removed "observational instrument".
--------------------------------
Reviewer’ note: Many information about instruments of evaluation is offered, however, a lack of information is showed about teaching/learning methodologies. I think the introduction should have a great modification in order to offer more insight into learning strategies focused on technical or tactical methods. This change implies a great modification of this section. Moreover, I think the methodological information (pre-post tests) is not necessary here. The reader can find all the details in the next sections.
Authors’ response: The introduction was structured as follows: i) introduction of the study topic; ii) description of the main characteristics of the DI and TGA methodologies; iii) previous researches that comparing methodologies in school and the results obtained; iv) it was mentioned that teachers only assess the technique in the Physical Education classes and they forget about the tactic. Therefore, various instruments were cited to evaluate the tactic in different invasion sports (e. g. football), since their teaching is necessary; v) objectives and hypotheses of the study. We consider that the introduction follows a correct structure.
--------------------------------
Reviewer’ note: Please, give more information about the panel of experts (experience, level, number…). I think it is a great contribution to this research study.
Authors’ response: More information about the expert panel was added (Line 150-151).
--------------------------------
Reviewer’ note: Please, explain with more detail what the authors consider as “student experience” (lines 146-147). Experience in deliberate practice? in competition?
Authors’ response: More detail what the authors consider as “student experience” were added (Line 152 to 153).
--------------------------------
Reviewer’ note: I think this information could be set close to the definitions of the variables (after line 165). Moreover, I suggest including more details related to the categories: 1 = inadequate, 2 = neutral, and 3 = adequate. I encourage explaining it according to the different components (DM, TE, and FR).
Authors’ response: More details related to the categories were included: 1 = inadequate, 2 = neutral and 3 = adequate (Line 185 to 188). Citation was added that specifying the criteria [29].
García-Ceberino, J.M.; Antúnez, A.; Ibáñez, S.J.; Feu, S. Design and Validation of the Instrument for the Measurement of Learning and Performance in Football. Int. J. Environ. Res. Public Health 2020, 17, 4629, doi:10.3390/ijerph17134629.
--------------------------------
Reviewer’ note: Regarding the information shown in lines 201-204, please, give more details about the information requested to the students and how this information was managed in order to create properly balanced teams.
Authors’ response: More information about create properly balanced teams was added (Line 123).
--------------------------------
Reviewer’ note: Lines 208. Four learning situations were used, but how many were “simple” vs “complex” tasks? Please, also offer a greater explanation about the key instructions (feed-forward) used by the teacher before each exercise. The information about TGAS and DIS sessions is excessively general (like an introduction). Please, give more details about the task constraints, specific goals for sessions, etc. (no only about the general aims).
Authors’ response: More learning situations were used (…). A citation [32] was added that explains the structure and organization of the intervention programs (contents, objectives, feedbacks, sessions, tasks, etc.) (Line 223).
García-Ceberino, J.M.; Feu, S.; Ibáñez, S.J. Comparative Study of Two Intervention Programmes for Teaching Soccer to School-Age Students. Sports 2019, 7, 74, doi:10.3390/sports7030074.
--------------------------------
Reviewer’ note: In this “procedure” sub-section, please, offer more details about the pre-post test situation. Finally, it was done an inter-reliability process?
Authors’ response: Intra-rater reliability was calculated because the 3vs3 matches were encoded by a single coder (Line 230 to 241). Likewise, in line 176 to 177, inter-rater reliability of the observational instrument was mentioned.
--------------------------------
Reviewer’ note: Concerning the 2 “statistical analysis” sub-section, please, add the level of significance used in the tests.
Authors’ response: Level of significance was added (Line 264).
--------------------------------
Reviewer’ note: Lines 313-onwards. According to table 3, a few statistical differences were found between groups in the pre-test. This fact should be commented on.
Authors’ response: This fact was commented (Line 323 to 324).
Reviewer 2 Report
Summary of the research and overall impression:
The authors assessed two instructional methods in a parallel design in 5th graders at a school setting on the outcome of football play efficacy. Inclusion criteria demanded at least 80% of lesson attendance for data to be included in analysis.
The reviewer commends the authors for addressing the analysis of teaching strategies in a child population. The methodology and design of the study are sound and well explained. Specifically, the depiction of instrument validity and assessment methodology follows a clear rationale.
The statistical analyses performed are rigorous and accurate, and the reviewer would like to thank the authors for spending ample time in writing out a detailed analysis approach, rather than a simple blanket statement. More precisely, the identification of potent covariates and effect modifiers, such as play experience adds to the overall importance of this manuscript. Well done.
Only minor issues should be addressed:
Line 103-108: "...A pre-test/post-test design was used and the study was conducted within a typical educational setting with instruction provided in the sport of football. The groups underwent different instructional protocols with Group A following a program based on TGA principles and Group B adhering to a program that utilized the DI method. Previous student experience with the game of football was also examined as a possible influence upon learning." This paragraph seems to be more suited for the Materials and Methods section i.e. "2.1 Design of the Study". Please adjust and merge the listing of hypotheses behind the purpose statement.
Line 200 - 200: The use of a questionnaire to even out the teams for assessment is a reasonable approach. Though the authors asked for previous playing experience, was the motivation for learning the sport of football also assessed? And could this also have an impact? A brief statement in the discussion will add to the overall merit of the manuscript.
The reviewer is looking forward to seeing the revised version of the manuscript and again would like to congratulate the authors on an excellent report.
Author Response
First of all, we would like to express our gratitude to reviewer 2 for the time in reviewing our manuscript and for providing us comments helpful to improve this manuscript quality. We have found suggestions very constructive and have answered their concerns point by point.
--------------------------------
All manuscript
- A native translator performed a grammatical revision of the manuscript.
- All corrections were marked in red.
--------------------------------
Reviewer’ note: Line 103-108: "...A pre-test/post-test design was used and the study was conducted within a typical educational setting with instruction provided in the sport of football. The groups underwent different instructional protocols with Group A following a program based on TGA principles and Group B adhering to a program that utilized the DI method. Previous student experience with the game of football was also examined as a possible influence upon learning." This paragraph seems to be more suited for the Materials and Methods section i.e. "2.1 Design of the Study". Please adjust and merge the listing of hypotheses behind the purpose statement.
Authors’ response: Line 103 to 108 were added to the “2.1 Design of the Study” section. Likewise, the listing of hypotheses behind the purpose statement was adjusted (Line 115 to 119).
--------------------------------
Reviewer’ note: Line 200 - 200: The use of a questionnaire to even out the teams for assessment is a reasonable approach. Though the authors asked for previous playing experience, was the motivation for learning the sport of football also assessed? And could this also have an impact? A brief statement in the discussion will add to the overall merit of the manuscript.
Authors’ response: The influence of motivation for learning the sport of football was indicated in the discussion. In this sense, a greater perceived competence favors the intrinsic motivation and it causes students have a greater interest in sports practice (Line 334 to 338).
Reviewer 3 Report
I am happy to have had the opportunity to review this manuscript and believe the article is well done. However, I do not believe this is the correct journal outlet for the manuscript. There is no attempt to tie any of the research to any discussion of sustainability and, in fact, I do not want to suggest that the authors alter the manuscript to placate the journal readership. Rather, I suggest the authors find a better outlet that focuses on physical education, sport, coaching, etcetera.
I believe if I shared my comments with you, it would be to shift the paper to have a sustainability focus, but I don't think that is where the paper should go. I think the paper is good and would be better served in a different outlet.
So, you will see that I rejected your paper, but that is only based on the fact that I don't believe it is good for this journal. If the editors disagree, I will send you the review, but I don't think that is what is best for this good research article.
Author Response
First of all, we would like to express our gratitude to reviewer 3 for the time in reviewing our manuscript.
--------------------------------
All manuscript
- A native translator performed a grammatical revision of the manuscript.
- All corrections were marked in red.
--------------------------------
Reviewer’ note: I do not believe this is the correct journal outlet for the manuscript. There is no attempt to tie any of the research to any discussion of sustainability and, in fact, I do not want to suggest that the authors alter the manuscript to placate the journal readership. Rather, I suggest the authors find a better outlet that focuses on physical education, sport, coaching, etcetera.
Authors’ response: This manuscript was uploaded to a special issue "Advances in Physical Education, Exercise and Sport: Towards a Healthy and Sustainable Lifestyle" (link: https://www.mdpi.com/journal/sustainability/special_issues/phy_edu_life_sus). In this sense, we consider that it appropriate for this special issue because it shows several strategies to promote the sports practice of the students and favor healthy lifestyles (e.g. Line 334 to 338).
Reviewer 4 Report
The manuscript is well conducted but the idea is not really new. The introduction can be improved by indicating aspects of previous studies regarding the inclusion of female subjects in this kind of approach.
Author Response
First of all, we would like to express our gratitude to reviewer 4 for the time in reviewing our manuscript and for providing us comments helpful to improve this manuscript quality. We have found suggestions very constructive and have answered their concerns.
--------------------------------
All manuscript
- A native translator performed a grammatical revision of the manuscript.
- All corrections were marked in red.
--------------------------------
Reviewer’ note: The manuscript is well conducted but the idea is not really new. The introduction can be improved by indicating aspects of previous studies regarding the inclusion of female subjects in this kind of approach.
Authors’ response: A paragraph was added to mention the involvement of the female in technical and tactical learning. In addition, previous studies regarding the inclusion of female were cited (Line 78 to 81).
González-Espinosa, S.; Mancha-Triguero, D.; García-Santos, D.; Feu, S.; Ibáñez, S.J. Diferencia en el aprendizaje del baloncesto según el género y metodología de enseñanza. Revista de Psicología del Deporte 2019, 28, 86-92.
Mesquita, I.; Farias, C.; Hastie, P. The impact of a hybrid Sport Education-Invasion Games Competence Model soccer unit on students' decision making, skill execution and overall game performance. European Physical Education Review 2012, 18, 205-219, doi:10.1177/1356336x12440027.